# The Immune Escape Mechanisms of *Mycobacterium Tuberculosis*

**DOI:** 10.3390/ijms20020340

**Published:** 2019-01-15

**Authors:** Weijie Zhai, Fengjuan Wu, Yiyuan Zhang, Yurong Fu, Zhijun Liu

**Affiliations:** 1School of Clinical Medicine, Weifang Medical University, Weifang 261053, China; weijie.zhai799@outlook.com (W.Z.); wfj5678@outlook.com (F.W.); zhangyiyuan912@gmail.com (Y.Z.); 2Department of Medical Microbiology, Weifang Medical University, Weifang 261053, China; yifuyurong@163.com

**Keywords:** *Mycobacterium tuberculosis*, macrophages, apoptosis, immunology, immune escape

## Abstract

Epidemiological data from the Center of Disease Control (CDC) and the World Health Organization (WHO) statistics in 2017 show that 10.0 million people around the world became sick with tuberculosis. *Mycobacterium tuberculosis* (MTB) is an intracellular parasite that mainly attacks macrophages and inhibits their apoptosis. It can become a long-term infection in humans, causing a series of pathological changes and clinical manifestations. In this review, we summarize innate immunity including the inhibition of antioxidants, the maturation and acidification of phagolysosomes and especially the apoptosis and autophagy of macrophages. Besides, we also elaborate on the adaptive immune response and the formation of granulomas. A thorough understanding of these escape mechanisms is of major importance for the prevention, diagnosis and treatment of tuberculosis.

## 1. Introduction

Tuberculosis (TB) is an infectious respiratory disease that seriously endangers health. CDC and WHO statistics in 2017 show that 10.0 million people around the world became infected with TB and there were 1.3–1.6 million TB-related deaths. Despite the availability of a vaccine for nearly 100 years, the global infection rate of *Mycobacterium tuberculosis* (MTB) is still approximately one in three people [1,2]. However, the pathogen is eliminated in only 10% of people, while it can escape during the process of infection and remain dormant in old lesions, so the infection is hard to control. A decline in immunity may lead to the resuscitation of MTB in response to perturbations of the immune response and active tuberculosis ensues [3]. The immune response after MTB infection is complex and the bacteria have intricate immune escape mechanisms. Therefore, it is particularly important to clarify these mechanisms for the effective diagnosis and treatment of pulmonary tuberculosis. In this article, we review the escape mechanisms of MTB.

MTB is the pathogen that causes tuberculosis. After a host is infected with MTB, there is a 10% probability that he/she will develop active tuberculosis and the bacterium may invade multiple organs [4]. Healthy individuals can be infected via the respiratory tract, the digestive tract, damaged skin and mucous membranes [5]. Inhalation of droplets containing MTB is the main route of infection. Complex bacterial cell membranes contain methyl branched-chain fatty acids that protect them from host enzymes and enable them to escape immune responses [4]. There are various aspects of macrophage-mycobacterium interactions, such as the binding of MTB to macrophages via surface receptors [6], phagosome-lysosome fusion [7], mycobacterial growth, inhibition/damage through free-radical mechanisms (reactive oxygen and nitrogen intermediates) [8], cytokine-mediated mechanisms to recruit accessory immune cells for a local inflammatory response and the presentation of antigens to T cells for the development of acquired immunity [9].

When MTB enters the body, it can initiate a series of immune responses by interacting with macrophages. Receptors such as Toll-like receptors (TLRs) reside on macrophages and interactions between MTB and the various TLRs are complex, distinct mycobacterial components appearing to interact with different members of the TLR family [10]. Sonia Davila et al. noted that TLR8 protein is expressed directly in differentiated macrophages upon MTB infection [11]; Nod-like receptors [12], comprise a large family of intracellular pattern recognition receptors that are characterized by the presence of a conserved NOD [13]; genetic variations in Nod-like receptors are associated with the development of inflammatory disorders [14]; and C-type lectin-like receptors [15] also play a role in mycobacterial recognition through regulating mycobacterial-induced cytokine production by cells like DCs. So, some mechanisms that block the maturation and acidification which normally present antigen to T lymphocytes to release lymphokines that activate macrophages through positive feedback, enhance their maturation and acidification and accelerate the mechanism for killing intracellular pathogens in autophagic lysosomes.

Therefore, some mechanisms that block the maturation and acidification of lysosomes can help MTB escape immunity [16,17].

## 2. *M. tuberculosis* Inhibits the Maturation and Acidification of Phagolysosomes

### 2.1. M. tuberculosis Inhibits the Maturation of Phagolysosomes

The structural characteristics of MTB and the composition of its mycelium inhibit the maturation and acidification of the phagolysosome. Bacterial proteins such as early secretory antigen-6/culture filtrate protein and ATP1/2 (secretion ATPase1/2, secreted secA1/2 protein) decrease the pH by preventing the accumulation of vacuolar ATP and GTP enzymes and this affects phagocyte maturation. Another protein, initially termed tryptophan aspartate rich coat protein and now known as coronin 1, is recruited to phagosomes containing active bacilli but is rapidly released from phagosomes containing inactive mycobacteria [18]. MTB inhibits the formation of lysosomes by raising the coronin 1 expression on the host phagocyte membrane; the duration of the recruitment process and the amount of coronin 1 are positively correlated with the amount and activity of activated MTB in the microsomes [19,20]. Besides, the cytokine interferon (IFN)-α inhibits maturation by inducing the production of interleukin (IL)-10 in a STAT1-dependent manner. IL-10 acts to reduce excess IL-1β and thus suppresses the caspase1-dependent IL-1β maturation of pleural fluid mononuclear cells [21]. BCG live vaccine can prevent the maturation of phagosomes by blocking expression of the lysosomal glycoprotein LAMP-1 in the phagosome exon [19]. PKnG is a protein similar to protein kinase in eukaryotes. On the one hand, PKnG enhances MTB metabolism, growth rate, virulence and drug resistance by reducing the expression of GlpK and ALD, increasing the expression of Ag85A and Ag85C, inhibiting the maturation of lysosomes and enhancing the infectivity of bacteria. On the other hand, PKnG secreted by MTB prevents the fusion of phagosomes and lysosomes by enhancing signal transduction in host cells. Therefore, there is an interaction between MTB and PKnG [22,23,24].

Inhibiting the fusion of phagosomes with lysosomes is also an important mechanism of inhibiting the maturation of phagosomes/lysosomes in macrophages. Studies have shown that the pro-inflammatory transcription factor NF-κB (nuclear factor kappa B) regulates the release of lysosomal enzymes into phagosomes, thereby regulating the killing of pathogens. Furthermore, the production of membrane transport molecules is increased by NF-κB, regulating the fusion of phagolysosome fusion during infection [7]. Phosphatidylinositol 3-phosphate (PI3P) is an important component of the macrophage cell membrane located on the early endosome and phagosome surface. After infection by MTB, lower biosynthesis [25] and via trafficking the toxin lipoarabinomannan (LAM) along with the calmodulin-dependent production of PI3P [26] suppresses the process of fusion between phagosomes and lysosomes. In considering dendritic cells (DCs) as host cells for MTB, DC-specific intercellular adhesion molecule-3 grabbing nonintegrin (DC-SIGN) captures and internalizes intact MTB through the mycobacterial cell-wall component ManLAM, which is also secreted by macrophages infected with MTB. Strikingly, the combination of DC-SIGN with ManLAM results in the inhibition of DC maturation [27,28]. As noted above, MTB can survive in macrophages by inhibiting lysosome-phagosome fusion (Figure 1). However, some MTB strains cannot adapt to subsistence within the endocytic vesicles of macrophages [29]. These bacteria can create favorable conditions for survival in the cytoplasm by activating phospholipase A2 in host cells [29].

### 2.2. M. tuberculosis Inhibits the Acidification of Phagolysosomes

MTB inhibits the maturation of phagocytosis by suppressing the acidification of phagosomes and then persists in the relatively lower acidic environment (pH~6.2) [30]. First, MTB inhibits phagosome acidification by changing its composition; the structure and specific molecules on the cell wall serve as a barrier, allowing the macrophages to maintain a neutral pH [30]. Second, the protein tyrosine kinase A (PtkA) is encoded in the same operon as PtpA. In the investigation by Tsui et al., a PtkA deletion mutant MTB was unable to survive in the THP-1 macrophage infection model, indicating an inability of the mutant to inhibit phagosome acidification, showing that PtkA plays a direct role in the acid inhibition [31]. Besides, infection of macrophages with MTB leads to the secretion of granulocyte-macrophage colony-stimulating factor, triggering the expression of cytokine-inducible SH2-containing protein (CISH) through mediation by STAT5. V-ATPase catalytic subunit A can also ubiquitinate and degrade proteasomes by producing CISH [32]. IFN-γ-inducible nitric oxide synthase 2 directly activates the infected macrophages and then suppresses the replication of intracellular MTB. A member of the 47-kD guanosine triphosphate family, LRG-47, acts independently of nitric oxide synthase 2 to protect macrophages against disease. Macrophages lacking LRG-47 cannot be completely acidified, thereby reducing their immune response to MTB [33].

## 3. *M. tuberculosis* Inhibits Oxidative Stress and the Function of Reactive Oxygen and Reactive Nitrogen Intermediates

Oxidative stress is a disorder of pro/antioxidant balance, resulting in potential damage. Oxidative stress can damage DNA bases and induce protein oxidation and lipid peroxidation. Underlying the prolonged MTB latency in the host is not only the inhibition of macrophage phagocytosis, lysosome maturation and acidification but also the inhibition of oxidative stress. SigH, one of 12 MTB alternative sigma factors, is induced by heat, oxidative and nitric oxide stresses [34]. Dutta et al. have found that SigH is required for long-term infection and a SigH-dependent regulon modulates the interaction of host innate immune responses with MTB [35]. The interrelationship between the MTB gene cluster Rv0014c-Rv0019c [PknA (encoded by Rv0014c) and FtsZ-interacting protein A (FipA) (encoded by Rv0019c)] and FtsZ and FtsQ, from the division cell wall cluster also contribute to the escape of MTB from oxidative stress. FipA that interacts with FtsZ and FtsQ establishes the PknA-dependent phosphorylation of FipA at T77 and FtsZ AT T343 is needed for cell division under oxidative stress. An MTB strain with FipA disruption is less able to survive in an oxidative environment [36]. Mycothiol (MSH; acetyl-Cys-GlcN-Ins) is a major thiol in MTB. It has antioxidant activity and detoxifies various compounds to help MTB remain in macrophages. One gene in the biosynthesis of MSH is mshD, which functions in encoding mycothiol synthase, the final enzyme in MSH biosynthesis. An MTB strain with knockout of the mshD gene has heightened sensitivity to H_2_O_2_ and grows poorly on agar plates lacking catalase and oleic acid [37]. Wag31 is a MTB protein belonging to the divIVA family. wag31 first binds amino-acid residues in SET domain through nuclear receptors and then interacts with penicillin-binding proteins. And it can protect MTB from oxidative stress-induced cleavage [38]. Interestingly a DivIVA homolog Mup012c is located on virup in protecting M. ulcerans during oxidative stress [39].

With the excitation of phagocytosis, macrophages produce a respiratory burst and generate reactive oxygen and nitrogen intermediates. Similarly, polymorphonuclear neutrophils (PMNs) are key components in the first line of defense against MTB. PMNs defend against pathogens by eliminating them and generating reactive oxygen species (ROS). Two prevalent MTB lineages in Argentina, Latin America and the Mediterranean, independent of the ability to enter PMNs, induce high levels of PMN apoptosis by triggering signaling mechanisms that involve ROS generation via p38 activation, leading to enhanced effector actions [40]. Also, the thicker cell wall of MTB and its specific component phospholipase D effectively block the action of ROS [19]. Imai et al. [41] found that in the presence of Mn^+^, MTB is able to resist the toxic effects of reactive oxygen and nitrogen intermediates because of its peroxidase activity via the production of catalase peroxidase. The KatG and TrxB2 enzymes of MTB help to resist an oxidative environment because their corresponding genes are significantly increased by H_2_O_2_ and NO, indicating that these two enzymes provide resistance to an oxidative environment [8]. Through the inhibition lipopolysaccharide-induced NF-κB-dependent gene expression via decreased synthesis of ROS, the filtrate protein CFP-10 and the early-secreted antigenic target ESAT-6 of the RD-1 region of the MTB genome inhibit oxygen free radicals. Furthermore, the complex of CFP-10 and ESAT-6 is more efficient than each protein alone [42]. Lsr2, an MTB protein with histone-like features, only protects MTB from reactive oxygen intermediates but not reactive nitrogen intermediates [43]. Shin et al. [44] have shown that by regulating the production of ROS via the c-Jun N-terminal kinase-ROS signaling pathway, the EIS gene also helps MTB to survive oxidative stress. MTB cannot synthesize glutathione; instead, it synthesizes two major low molecular weight thiols, MSH and ergothioneine (ERG). Gamma-glutamylcysteine (GGC) is an intermediate in glutathione and ERG synthesis. Five genes (egtA, egtB, egtC, egtD and egtE) are involved in the synthesis of ERG; however, the ERG-deficient ΔegtB mutant which accumulates GGC resists oxidative and nitrosative stress more strongly than the ERG-deficient and GGC-deficient ΔegtA mutant. This finding suggests that GGC is involved in the detoxification of reactive oxygen and nitrogen species in MTB [45].

## 4. *M. tuberculosis* Inhibits Apoptosis and Autophagy

The apoptosis of host cells is mainly related to the virulence of MTB [46]. Riendau et al. found that the weakly toxic MTBH37R strain and the BCG strain of *M. bovis* induce intense apoptosis of THP-1 cells differentiated by phorbol myristate acetate as a model of in vitro infection in which to study the mechanisms by which MTB induces macrophage apoptosis and their interaction [47]. In contrast, the wild-type MTB virulence strain H37Rv and *M. bovis* cannot strongly induce the apoptosis of macrophages. This indicated that the virulence components of MTB strains such as miR-30A regulate the apoptotic response of macrophages [48]. Overexpression of miR-30A suppresses the elimination of intracellular MTB and this is achieved by inhibiting autophagy [30]. Invasion by MTB and host cell apoptosis require cellular factors, signaling proteins and regulation of the pathways involving TNF-α, IFN-γ, transforming growth factors and IL-6, IL-12, IL-4 and IL-10 [49,50]. NKT cells can produce and release IFN-γ to inhibit the growth of MTB in macrophages [51,52,53,54]. DCs and macrophages are the main components of the first line of defense against MTB and they can also maintain the complementary function of eliminating infectious bacteria [55,56,57]. Surprisingly, infection by MTB is mainly based on cellular immunity, while the role of humoral immunity is controversial [58,59]. When exposed to MTB, T cells are activated as a basic component of the protective response [60]. Th1 cells secrete IL-2 and IFN-γ and play a protective role in intracellular infections. Th2 cells secrete IL-4, IL-5 and IL-10, which have negative effects on the immune response [61]. By comparison, the proportions of T cells expressing γ/δ T cell receptors from tuberculosis patients and controls are similar, so it is clear that the γ/δ cells play a role in the early immune response [62]. In general, macrophages infected by MTB have 3 outcomes: necrosis, apoptosis or survival [63]. At the early phase of infection, apoptosis is a major defense mechanism of macrophages against MTB [64]. The apoptotic cells present antigen to DCs to enhance acquired immunity. Apoptosis controls the reproduction of bacteria and reduces their viability in cells. Autophagy is regulated by associated genes [65]. It is a homeostatic process that is responsible for removing unnecessary materials and degrading nonfunctional cytoplasmic components (proteins, lipids and organelles) via lysosomes [66,67,68]. Meanwhile, as a vital immune defense mechanism, autophagy also participates in inherent immunity and the adaptive immune response [69]. The apoptosis of TNF-induced macrophages is similar to that induced by MTB but the cell death induced by non-apoptotic complement has no effect on the activity of bacteria [70]. Thus, the inhibition of apoptosis and autophagy by strongly virulent MTB can lead to immune escape and/or latent infection [71].

### 4.1. Weakly Virulent M. tuberculosis Promotes Apoptosis

MTB cord factor and sodium sulfate contribute to the apoptosis of macrophages and induce them to produce a protective immune response against infection. MTB modulates inflammation at distinct stages [72]. After infection with MTB, TLRs activate mitogen-activated protein kinase to induce the production of pro-inflammatory factors like monocyte chemoattractant protein 1, TNF-α and IL-6. Patients infected by weakly virulent MTB can compensate for the loss of IL-2 to a certain extent [73,74]. P19 lipoprotein on the cell wall of attenuated MTB stimulates the differentiation of CD4^+^ cells and also promotes the release of IL-6, IFN-gamma, IL-12 and other cytokines [75]. The studies of Behar et al., have shown that P19 may be the apoptosis-inducing factor of weakly virulent MTB [76]. Its polypeptide structure on the molecular surface can signal through TLR-2 and activate caspase-8, resulting in apoptosis; however, this mechanism is time- and dose-dependent [77,78]. Furthermore, P19 can serve as a mannose receptor on the surface of macrophages and this promotes the phagocytosis of weakly virulent MTB by macrophages [79].

### 4.2. Strongly Virulent M. tuberculosis Inhibits the Apoptosis of Macrophages

Some proteins secreted by strongly virulent MTB, such as superoxide dismutase, hydrogen peroxide/peroxidase KatG, serine threonine protein kinase PknE, I type NADH dehydrogenase NuoG, Rv3654c and Rv3655c inhibit macrophage apoptosis [80,81]. They act by regulating the production of NO and pro-inflammatory cytokines to change the relationship with TLRs. Therefore, they inhibit TNF-α-induced apoptosis and MTB interferes with the caspase [82], JAK2/STAT1 [83], TNF-α [84] and Bcl-2 pathways to reduce macrophage apoptosis and increase the survival rate of pathogens [85]. The coexistence of MTB and the host is achieved through the above mechanisms. The MTB early secretory proteins CFP-10 and ESAT6 regulate macrophage apoptosis at different stages of bacterial infection through regulating TNF-α [86]. In addition, MTB stimulates the release of IL-10 by immune cells such as monocyte macrophages, B cells, cytotoxic T cells and NK cells and increases the release of TNFR2, thereby blocking the activation of TNF-α. The transmembrane protein Bcl 2 plays an anti-apoptotic role by controlling the transport of intracellular and extracellular substances, inhibiting Ca^2+^ release, and/or blocking the accumulation of intracellular peroxide [87].

## 5. The Effects of Iron, Hydrogen and Calcium Ions in Macrophages

### 5.1. Iron Ions Inhibit Lysosome Formation

The growth of MTB requires it to take up exogenous iron [88]; this is accomplished by the activation of Rab5 and via mediation of the transferrin receptor. Iron is indispensable as an auxiliary factor in the action of MTB-encoding enzyme; it is involved in electron transport and oxidative metabolism; and it is one of the elements required for the synthesis of amino-acids, pyrimidine nucleotides and a series of nutritional and genetic substances [89]. Clinical practice has shown that the level of iron intake in tuberculosis patients is proportional to the risk of death [90]. Consequently, host cells can prevent the growth of MTB by limiting their iron intake [91]. Natural resistance-related membrane protein 1 (Nramp1) [56] is a membrane protein used to resist invasion by *Leishmania*, MTB and other pathogens. Inactivated macrophages in alveoli have weak antibacterial activity and cannot inhibit the growth of MTB. MTB can be transported to other places and present antigens, sensitizing the surrounding T lymphocytes. Sensitized lymphocytes produce multiple lymphokines, such as IL-2, IL-6 and INF-γ, whose interaction with TNF-α can kill MTB in the lesion [92]. IFN-γ is a primary lymphokine. Stimulation by IFN-γ increases the expression of Nramp1, which transports ferrous iron across the cell membrane. Therefore, one of the possible mechanisms for Nramp1 to resist the invasion of extracellular pathogenic bacteria is to pump iron out of the phagosome into the cytosol, thus limiting the access of MTB to iron [93].

### 5.2. Hydrogen Ions Inhibit Lysosome Formation

Macrophages acidify phagosomes to reduce their pH to <5, which is not optimal for lysosomal enzyme activity but is most effective for killing pathogens [94]. As the main access of the pathogen is through the respiratory tract, alveolar macrophages are crucial defenders against pathogens [95]. Confocal scanning microscopy has revealed that in phagosomes that have contained live MTB for 6 h, the pH is >6, while phagosomes containing heat-killed MTB have a pH ≤4.5. Meanwhile, the phagosomes containing heat-killed MTB clearly fuse with lysosomes, while those containing live MTB are separated from lysosomes [96]. It is believed that early endosomes are membrane-bound organelles formed by endocytosis [97]. The proton-ATPase of the membrane maintains the acidity of endosomes by regulating the H^+^ concentration. When the H^+^ content in lysosomes is close to or reaches the required concentration, lysosomal zymogen is activated under the action of proton-ATPase, enabling its hydrolytic activity and its role in killing pathogens [98]. It has been shown that the proton-ATPase on the phagocytotic membrane lacks the proton pump, so extracellular H^+^ cannot be pumped into the phagosome [99], enabling MTB to survive there [2]. Macrophages activated by IFN-γ interfere with the uptake of nutrients by MTB so that the pH in the phagosome is maintained at ~5. The pH in macrophages that are not activated by IFN-γ is >6, so the number of such macrophages infected is greatly reduced. Therefore, loss of proton-ATPase may be the main reason for the absence of acidification in phagosomes containing MTB [100].

### 5.3. Calcium Ions Promote Lysosome Formation

Macrophages infected with MTB have a high demand for Ca^2+^. A change in the Ca^2+^ concentration is mainly caused by the activation of the purine receptors P2Y2 (purinergic receptor P2Y, G-protein-coupled, 2, P2RY2) and P2Y7 (purinergic receptor P2Y, G-protein-coupled, 7, P2RY7) on the surface of macrophages under the action of ATP [101]. P2Y2 instantaneously increases the intracellular Ca^2+^ concentration while P2Y7 is a pore-type receptor [102]. GTP (guanosine triphosphate) is an activator of P2Y2 but only works in synergy with P2Y7 [103]. Killing is achieved by strengthening the fusion of phagosomes with lysosomes. Increasing the extracellular Ca^2+^ concentration increases phospholipase D activity, which together promote the ATP-induced fusion of phagosomes with lysosomes [104]. It has been suggested that the loss of Ca^2+^ in macrophages containing MTB phagosomes reduces phagosome-lysosome fusion to some extent [105].

## 6. Other Mechanisms Helping *M. tuberculosis* to Escape Immune Responses

Before MTB is engulfed, selective pathways can be triggered by TLRs, among which, TLR2, TLR4 and TLR9 particularly activate the PI-3K pathway [106]. Moreover, numerous studies have shown roles of TLR1 [107], TLR2 [108], TLR4 [109], TLR6 [109] and TLR9 [110] in MTB recognition. A component of the MTB cell wall, LAM, activates TLR1 and TLR2. Besides, LAM initiates transmembrane signaling by altering the physical association between TLR1 and TLR2. TLR4 negatively regulates the degradation of phagocytes in macrophages through a pathway independent of NF-κB [111]. MTB has a special envelope composed of unique lipids, located at the host-pathogen surface and this contributes to the immune escape [112]. Chen et al. suggested that phagocyte MTB escapes into the cytoplasm to avoid lysosome killing. Antigenic mutation is also an important mechanism for avoiding killing; the emergence of structural mutations establishes an immune response by altering recognition and allows pathogens to persist [113]. MTB readily infects HIV patients [114] and HIV can also promote the progression of MTB infection [115]. The balance between T helper 17 (Th17) and T regulatory (Treg) cells plays a key role in maintaining normal immune function [116] but imbalance in the proportion of Th17 cells has been shown in patients with MTB and HIV co-infection [117].

Under normal conditions, MTB may target DC-SIGN to suppress cellular immune responses, since both immature and IL-10-treated DCs are not only less efficient in stimulating T cell responses but also induce a state of antigen-specific tolerance [118]. Cell-wall-related alpha-glucan can induce mononuclear cells to differentiate into DCS (Glu MODs), which have the same phenotype and functional behavior as DCS derived from MTB-infected mononuclear cells (Mt MoDCs). The difference is that Glu-MODCs cannot express CD1 or detect the presence of the lipid antigen from cloned CD1-restricted T cells. This inability of Glu-MoDCs allows MTB to evade both innate and acquired immune responses [119]. In vitro experiments on mouse bone marrow-derived DCs (BMDCs) and macrophages showed that their activation with IFN-γ and lipopolysaccharide inhibit the growth of intracellular bacteria in a nitric oxide synthase-dependent fashion. While this activation enables macrophages to kill intracellular MTB, the DCs cannot. Thus, the intracellular growth of MTB can be restricted to BMDCs and eliminated from macrophages. This suggests that DCs might serve as a reservoir for MTB in tissues like the lungs or lymph nodes [120]. The generation of CD83^+^ and CCR7^+^ DCs (Mt-MoDCs) requires the infection of monocytes with live MTB, since infection with heat-killed bacteria partially abrogates the effects on monocyte differentiation. It is clear that IL-4 and IFN-γ are ineffective in killing macrophages. Moreover, interference with the development of mononuclear cells into fully-competent DCS, as an escape mechanism, contributes to the intracellular persistence of MTB and avoids immune recognition [121].

MTB infection increases the expression of peroxisome proliferator-activated receptor gamma (PPARγ) via mechanisms including pattern recognition receptor activation, overexpression and activation resulting in increasing lipid droplet formation and downregulation of the macrophage response, suggesting that PPARγ expression aids MTB in circumventing the host response by acting as an escape mechanism [122]. As a secretory protein with immunogenic potency, MTB Rv1987 is encoded by region of difference-2 genes and modulates the host immune response towards a Th2 profile with lower secretion of IFN-γ but higher production of IL-4 and Rv1987-specific IgG antibody, which probably contribute to the immune evasion from host elimination [123]. Furthermore, the unique MTB antigen family Pro-Pro-Glu (PPE) is limited to the *Mycobacterium* genus, is prevalent among pathogenic *Mycobacterium* species and has been widely speculated to be a “molecular mantra” for escaping host immunity. It is closely associated with the ESAT-6 secretion system and largely located in the cell wall or cell membrane; all members of this family are characterized by a conserved N-terminal and a variable C-terminal. The expression of PPE protein is temporally modulated and highly expressed during MTB persistence [124]. One member of the PPE family is Pro-Pro-Glu 38 (PPE38), which inhibits the transcription of genes encoding MHC class I and decreases the number of CD8^+^ T cells in the spleen, liver and lungs. Another member of this family is the PPE68 protein, which is able to interact with and export several surface and secreted proteins [80].

It has been demonstrated that the MTB rv0431 gene, which encodes a cytoplasmic membrane junction protein with a special folding structure, also reduces the formation of MTB membrane vesicles rich in the macrophage TLR2 agonist LpqH, so this gene decreases interactions with macrophage receptors thereby suppressing immune stimulating factor production and reducing the killing of MTB [125]. Experiments on the upstream events responsible for the down-regulation of CD1 have shown that infection with live MTB decreases the steady-state CD1 mRNA level. The loss of CD1 on the cell surface is related to the ability of infected cells to deliver antigen to CD1-restricted T cells, so it can regulate the immune response processes associated with MTB [126]. MTB can also resist cytokine activity due to IL-1b and IL-8, growth-related oncogene-b and epithelial cell-derived neutrophil-activating peptide-78 and survive inside human macrophages [49].

## 7. Formation of Granulomas Help *M. tuberculosis* to Escape Immune Responses

The formation of granulomas is a primary host-defense mechanism for containing bacteria [127]. The components of granulomas are compact, comprising aggregates of immune cells, including lymphocytes on the outside surrounding blood-derived infected and uninfected macrophages, foamy macrophages, epithelioid cells and Langerhans cells [128] (Figure 2). More intriguing is that different forms of granuloma appear in different infectious states: classical granuloma is found in active disease and latent infection – in the center of this kind of granuloma is a necrotic area made up of dead macrophages and other cells [129]; non-necrotizing granulomas usually appear in active disease – these consist primarily of macrophages with a few lymphocytes, necrotic neutrophilic granulomas and fibrotic granulomas [130,131]; and fibrotic lesions mainly appear in latent TB but also in some active disease and are composed almost entirely of fibroblasts, with a minimal number of macrophages [129]. With respect to hypoxic granulomas, investigators have found that in latent infections MTB resides in the central hypoxic zone in a metabolically altered state but in active TB they can replicate in peripheral oxygenated areas [129].

The typical lesion of MTB infection is the granuloma, which is composed of macrophages, epithelioid cells, fibroblasts, lymphocytes and Langhans cells. They have an inherent inflammatory element but have evolved into a more complex and vital structure. When they play a role as a primary host-defense mechanism for containing bacteria, they also provide a shelter for MTB, some of which can live dormant in these structures for a long time until an opportunity arises for re-activation and spread. An understanding of the physiopathology and inflammatory status of granulomas is important for the treatment of tuberculosis.

While TLR2 functions as a defense system against MTB infection, it also plays a negative role. TLR2 immune responses can prolong the survival conditions for MTB in macrophages. TLR2 increases the synthesis of inflammatory cytokines in macrophages, accumulates immune cells and lead to the formation of granulomas [132]. According to the hypotheses of Ehlers et al., latent MTB in granulomas can still regulate the immune response [133]. Within the mature granuloma, TNF-α-derived signals recruit highly-dynamic effector T cells and maintain the granuloma structure [134]. From MHC compartments, peptide-loaded MHC class II molecules function in activating T cells in order to generate an immune response against MTB and this action takes place in the plasma membrane [135]. Moreover, in an immune microenvironment, fibroblasts express MHC class II to Present antigens to CD4^+^ T cells after being activated by IFN-γ. Besides, IFN-γ-treated fibroblasts present antigens as well as peptides and isolated proteins. In contrast, MTB-infected fibroblasts are deficit in antigen presentation, indicating that MTB can escape T helper immune surveillance via infecting fibroblasts [136]. Another responsible factor, PE_PGRS47 protein of MTB (expressed by the Rv2741 gene) inhibits MHC class II-restricted antigen presentation by MTB-infected DCs, thus suppressing autophagy and helping MTB escape from the killing effects of innate and adaptive immunity [137] (Figure 3).

## 8. Pre-clinical Animal Models

A suitable animal model is critical to evaluate drug efficacy against MTB. The mouse is by far the most practical and widely-used animal model for drug discovery but it does not recapitulate human pathology. Especially, the granulomas are poorly organized and lack necrosis, fibrosis or hypoxia [138]. A rabbit model can be used to study latent MTB which is characterized by persistent, host-contained infection that can be reactivated on immunosuppression [139,140]. Using non-human primates is undoubtedly the most costly and resource-intensive model, since these are the closest experimental models to humans [141]. This model is currently being validated with standard anti-TB drugs and gene-expression studies to characterize the metabolic state of MTB. Combining the non-human primate model with conditional gene-silencing [142] techniques provides the first realistic experimental strategy for validation of drug targets for the treatment of TB. Although there are still some gaps like time, cost and compound requirements, when a candidate drug is about to enter full clinical development, it warrants efficacy studies in the non-human primate model [129].

## 9. Conclusions

To date, the understanding of the mechanism of immune escape by MTB remains limited. Over the last fifty years, numerous studies have investigated the pathogenic mechanisms of MTB and the immune response. However, tuberculosis caused by MTB still endangers health worldwide. This is mainly due to its immune escape mechanisms, which greatly enhance its survival in the host. Aspects of MTB infection and the immune escape mechanisms provide a basis for the future treatment of tuberculosis. By suppressing macrophage maturation and lysosomal acidification as well as inhibiting oxidative stress, apoptosis and autophagy, MTB is capable of remaining latent in the host. Besides, iron, Ca^2+^ and H^+^ also function in the immune escape of MTB. This process is achieved via various proteins and genes. Therefore, this work opens up a new avenue for drug research to treat tuberculosis.

## Figures and Tables

**Figure 1 ijms-20-00340-f001:**
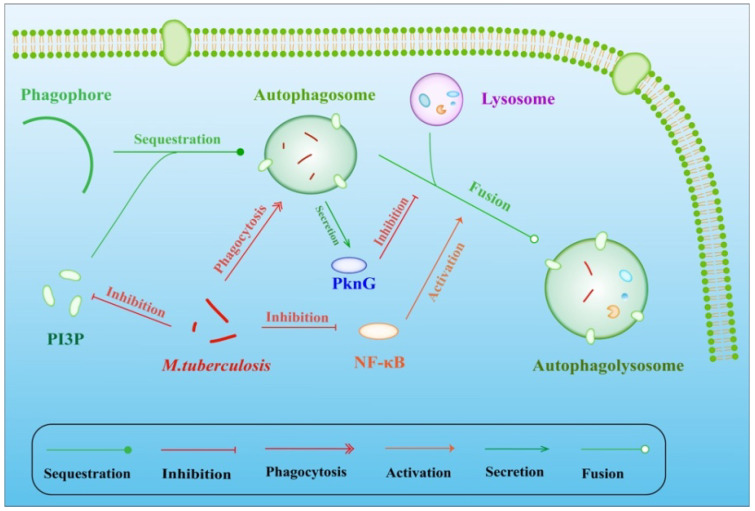
MTB evasion by inhibiting fusion of lysosomes with phagosomes. While the secretion of PknG by phagocytic MTB directly inhibits the fusion of phagosomes with lysosomes, the suppression of NF-κB also decreases this fusion. As an important component on the phagosome surface, lower biosynthesis and higher hydrolysis of PI3P also suppresses the fusion, providing an escape channel for MTB.

**Figure 2 ijms-20-00340-f002:**
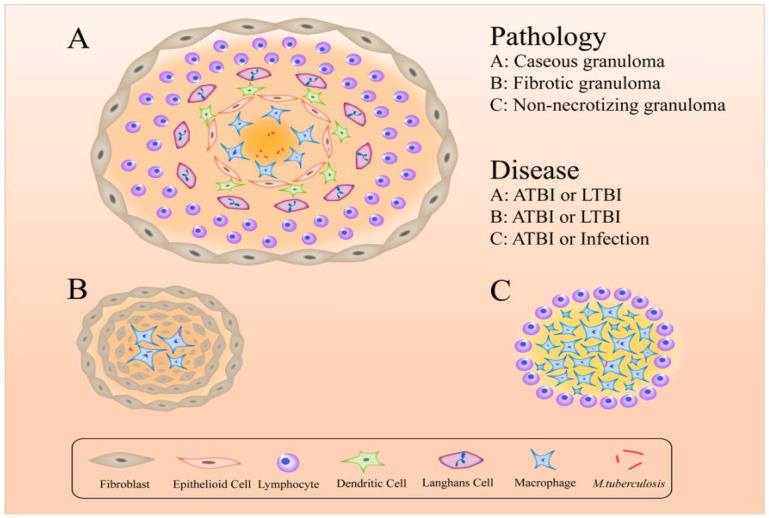
Active (ATBI), latent TB infection (LTBI) or infection granulomas that provide a shelter for bacteria.

**Figure 3 ijms-20-00340-f003:**
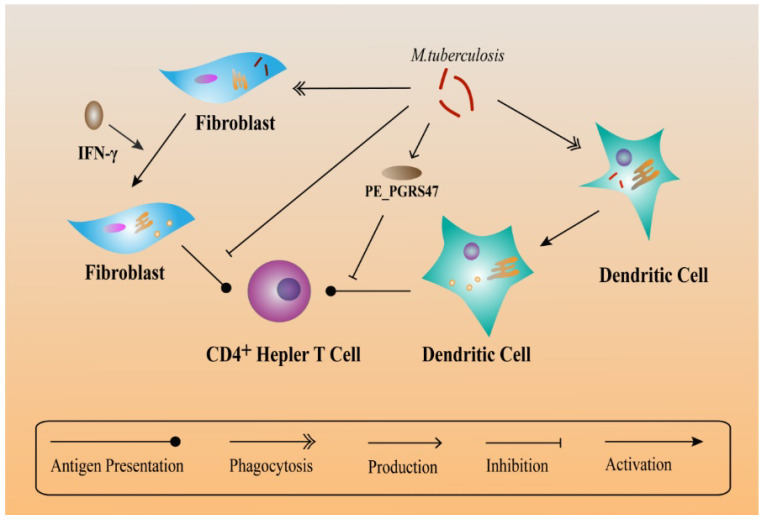
Relationship between MHC class II and *M. tuberculosis* evasion. As non-professional antigen-presenting cells, fibroblasts also present antigen derived from the processing of heat-killed MTB in addition to presenting peptides and isolated proteins. But MTB-infected fibroblasts are unable to present antigen from the bacteria. Peptide-loaded MHC class II molecules activate T cells to generate an immune response against MTB and this action occurs in the plasma membrane. However, PE_PGRS47 expressed by MTB can inhibit the presentation of MHC class II antigen.

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
