# Peer review of "The Immune Escape Mechanisms of *Mycobacterium Tuberculosis"

_ijms, 2019, doi:10.3390/ijms20020340_

Round 1
Reviewer 1 Report
Abstract: the focus of the study appears to be quite narrow, with the major focus being on the role of macrophages and apoptosis, autophagy. This does not justify the title of the review which portrays a more global phenomenon exhibited by Mtb in escaping immune responses upon infection. Either the title of the review should be changed to showcase the role of macrophages or the review abstract should include the innate immune responses and the adaptive immune responses as well.
1. Introduction: Referencing the latest WHO and CDC statistics is crucial in the first few sentences which is missing in the present form.
2. The sentence "A decline in immunity may lead to resuscitation of bacteria" is a very generic statement, it needs elaboration either in terms of TB specific resuscitation or if talking in general about bacteria, in terms of why and how this happens.
3. lines 41 - 47 should have more references, preferably one for each mechanism stated.
4. Lines 49- 50: Please mention the specific cell types producing these receptors, making it clear that these receptors are presented by a certain cell type is significant to elucidate the mechanism involved.
5. line 63: the term "inactive bacteria" should be changed.
6. line 86: Please mention which strains cannot adapt
7. Line 89: Please elaborate the survival conditions
8. Both the figure legend and Fig.1 need to be improvised considerably as it is too difficult to understand in its present form. The figure legend does not discuss all the components of the figure itself such as phagopore, also there should be a flow to the legend in the same manner as the figure so that it is very easy for the reader to follow the figure components. There seems to be a disconnect i the text and figure right now.
9. Lines 103-104: Does this statement pertain to therapeutics? If yes, then perhaps the statement must be made in a section pertaining to therapeutics. It appears a bit unwanted in this context.
10. There could be a mention about the sigH signalling in the context of oxidative stress.
11. There could be a mention about the role of TH17 in MTB, especially in HIV co-infected infections. In general, the co-infections and MDR, XDR topic has largely been left untouched, however, in a review such as this, it is inevitable that these should be discussed, if not in details, at least superficially in points here and there.
12. The heading 4.2 is not bold
13. Fig 2 could be made more interesting by comparing the granulomas in active, LTBI and co-infections. This would provide a far better picture of the differences in both structural components as well as immune factors present.
14. When discussing immune escape mechanisms in TB, it is also important to discuss a few non human primate studies, once again, to give a better understanding of the mechanisms. Since these would be the closest experimental models to humans.
Author Response
Point-by-point responses to the comments of the Editor and Reviewers, and a list of changes:
Reviewer 1:
Q: Abstract: the focus of the study appears to be quite narrow, with the major focus being on the role of macrophages and apoptosis, autophagy. This does not justify the title of the review which portrays a more global phenomenon exhibited by Mtb in escaping immune responses upon infection. Either the title of the review should be changed to showcase the role of macrophages or the review abstract should include the innate immune responses and the adaptive immune responses as well.
A: We thank this reviewer for the useful comments. Following your suggestion, the abstract of our manuscript has been revised and improved. Based on the content of the review, the innate immunity and adaptive immunity are summarized, and the structure of the whole article is more organized. Thank you again for your pertinent evaluation and careful guidance.
Q1: Introduction: Referencing the latest WHO and CDC statistics is crucial in the first few sentences which is missing in the present form.
A1: We have revisited the latest data content on the site and used it to replace the previous one.
Q2:The sentence "A decline in immunity may lead to resuscitation of bacteria" is a very generic statement, it needs elaboration either in terms of TB specific resuscitation or if talking in general about bacteria, in terms of why and how this happens.
A2: Thanks for the good suggestion. We have revisited the references and improved the relevant content.
Q3: lines 41 - 47 should have more references, preferably one for each mechanism stated.
A3: Thanks. We have added references and re-explained each mechanism, making each more understandable.
Q4: Lines 49- 50: Please mention the specific cell types producing these receptors, making it clear that these receptors are presented by a certain cell type is significant to elucidate the mechanism involved.
A4: We agree, and have added new references to make this part more accurate, as well as clarifying the correspondence between each cell type and its receptor; the related immune mechanisms have already been discussed.
Q5: line 63: the term "inactive bacteria" should be changed.
A5: Thanks for the suggestion; this has been corrected.
Q6: line 86: Please mention which strains cannot adapt
A6: Thanks for the suggestion. The manuscript has been revised as suggested and we describe some MTB strains lacking a mechanism involving activation of host cytosolic phospholipase A2 that cannot adapt to subsistence within the endocytic vesicles of macrophages.
Q7: Line 89: Please elaborate the survival conditions
A7: After checking more references, related content is now elaborated in the article.
Q8: Both the figure legend and Fig.1 need to be improvised considerably as it is too difficult to understand in its present form. The figure legend does not discuss all the components of the figure itself such as phagopore, also there should be a flow to the legend in the same manner as the figure so that it is very easy for the reader to follow the figure components. There seems to be a disconnect i the text and figure right now.
A8: We agree. In the revised paper, Fig. 1 has been improved to make the relevant content more clear and easier for readers to understand.
Q9: Lines 103-104: Does this statement pertain to therapeutics? If yes, then perhaps the statement must be made in a section pertaining to therapeutics. It appears a bit unwanted in this context.
A9: Thanks for the suggestion. We have decided to delete this to provide a more readable context.
Q10: There could be a mention about the sigH signalling in the context of oxidative stress.
A10: Yes, sigH is critical in long-term infection and plays a critical role in modulating the interaction of host innate immune responses with MTB. Appropriate content has been added in the text.
Q11: There could be a mention about the role of TH17 in MTB, especially in HIV co-infected infections. In general, the co-infections and MDR, XDR topic has largely been left untouched, however, in a review such as this, it is inevitable that these should be discussed, if not in details, at least superficially in points here and there.
A11: Thanks for your guidance. HIV and MTB can remain latent for a long time, and co-infection of HIV and MTB has a mutually reinforcing effect, accelerating the progression of the disease, in which Th17 plays an important role.
Q12: The heading 4.2 is not bold.
A12: Thanks. This has been changed.
Q13: Fig 2 could be made more interesting by comparing the granulomas in active, LTBI and co-infections. This would provide a far better picture of the differences in both structural components as well as immune factors present.
A13: That’s a very good suggestion. Comparing the pathological manifestations produced by different diseases can help the development of targeted treatments. In the revised paper, latent and acute TB infection are compared in terms of granuloma structure and hypoxic granulomas.
Q14:When discussing immune escape mechanisms in TB, it is also important to discuss a few non human primate studies, once again, to give a better understanding of the mechanisms. Since these would be the closest experimental models to humans.
A14: Thanks for the timely suggestion. We now compare three animal models, namely mice, rabbits, and non-human primates. There is no doubt that non-human primates are the best choice. The differences between them are clearly described in the revised paper.
Reviewer 2 Report
In the current manuscript the authors review the pathogenesis of tuberculosis. The topic is of interest but language is poor. There are formatting problems throughout the manuscript. The manuscript is not cohesive and remains largely vague. Although authors have referenced a large number of studies. I see many important recent articles missing. Th authors should add text box about important questions and important work done in last 5 years.
1. Line 15 “In addition, iron, calcium, and hydrogen ions also play crucial roles in the immune escape of MTB.” The role of Zinc is not discussed. Please explain if this is due to any specific reason.
2. Line 38 “Healthy individuals can be infected via the respiratory tract, the digestive tract, damaged skin, and mucous membranes.” Please cite appropriate Reference.
3. Line 42-45 “This is a matter of cardinal importance. There are various aspects of macrophage-mycobacterium interactions, such as the binding of MTB to macrophages via surface receptors, phagosome-lysosome fusion, mycobacterial growth, inhibition/damage through free-radical mechanisms (reactive oxygen and nitrogen intermediates), cytokine-mediated mechanisms to recruit accessory immune cells for a local…..” Sentence is not clear.
4. Line 51-“First of all, MTB invade macrophages, which normally present antigen to T lymphocytes to release lymphokines that activate macrophages through positive feedback, enhance their maturation and acidification, and accelerate the mechanism for killing intracellular pathogens in autophagic lysosomes. However, MTB escapes the immune response by inhibiting these mechanisms.” Please cite appropriate references (Shea and Siegel Clinical Immunology, Third Edition. PMID: 28137237, 27763255 ). Does the author mean Cytokine or lymphokine here ? Please be consistent with the terms.
5. The role of PknG is missing in section on maturation of phagolysosomes (PMID: 23084287, 26359969, 15155913)
6. Line 69 “The maturation of phagosomes can be arrested directly by live BCG vaccine, as phagosomes exibit only small amounts of the lysosomal glycoprotein LAMP-1 and processing of the lysosomal hydrolase cathepsin D is blocked)” This sentence is not clear.
7. Line 87 “Via a mechanism that activates host cytosolic phospholipase A2, these bacteria can promptly escape from the phagosome and create favorable conditions for survival in the cytoplasm of the host cell [17].” This sentence is not clear.
8. “Wag31, an MTB protein that belongs to the DivIVA family, protects MTB from oxidative stress induced cleavage due to its interaction with penicillin-binding protein through amino-acid residues in the nuclear receptor-binding SET domain [25].” The sentence is not clear.
9. The authors mention about Wag31 role. Interestingly a DivIVA homolog Mup012c is located on virulent plasmid of Mycobacterium ulcerans (PMID: 25412098). This protein may help in protecting M. ulcerans during oxidative stress. I think this can be mentioned.
10. Line 199 “The P19 lipoprotein on the cell wall of weakly virulent MTB stimulates the differentiation of CD4+ cells and the release of cytokines like IL-6, IFN-γ, and IL-12”. Does the author mean CD4 cells secrete IL-12 ? Please provide reference and rewrite the sentence.
11. Line 240 “This state is the most effective for killing parasitic pathogens, and the result is inhibition of phagosome-lysosome fusion[80] “ Reference is not correct. Please check.
12. Figure 3 legend:”and this action occurs in the plasma memrane However,”
13. Line 156. “ergothioneine (ERG). Gamma-glutamylcysteine (GGC) is an intermediate in
glutathione and ERG synthesis. Five genes (egtA, egtB, egtC, egtD, and egtE) are involved in
the synthesis of ERG;” Please see word fonts.
14. Line 168 “apoptosis of macrophages. .This indicated that
factors, and interleukins (IL-6, -12, -4, and -10)” Please see formatting.
15. Line 173 “DCS and macrophages are the main components of the first line of defense against MTB, “Please see formatting.
Author Response
Reviewer 2
Q: In the current manuscript the authors review the pathogenesis of tuberculosis. The topic is of interest but language is poor. There are formatting problems throughout the manuscript. The manuscript is not cohesive and remains largely vague. Although authors have referenced a large number of studies. I see many important recent articles missing. Th authors should add text box about important questions and important work done in last 5 years.
A: First of all, thank you for all the encouraging and critical comments. We have carefully considered them and revised the manuscript accordingly. With regard to language, we have carefully re-polished it and added descriptions of mechanisms. In order to enrich the content of our manuscript, we discuss more recent reports.
Q1: Line 15 “In addition, iron, calcium, and hydrogen ions also play crucial roles in the immune escape of MTB.” The role of Zinc is not discussed. Please explain if this is due to any specific reason.
A1: Thanks for this useful comment. The role of iron, hydrogen and calcium ions in immune the escape of MTB is described in detail, but we did not elaborate on the role of zinc ions in this process because it is still not clear.
Q2: Line 38 “Healthy individuals can be infected via the respiratory tract, the digestive tract, damaged skin, and mucous membranes.” Please cite appropriate Reference
A2: We are very grateful for the detailed guidance. The relevant references have been added.
Q3: Line 42-45 “This is a matter of cardinal importance. There are various aspects of macrophage-mycobacterium interactions, such as the binding of MTB to macrophages via surface receptors, phagosome-lysosome fusion, mycobacterial growth, inhibition/damage through free-radical mechanisms (reactive oxygen and nitrogen intermediates), cytokine-mediated mechanisms to recruit accessory immune cells for a local…..” Sentence is not clear.
A3: Thanks for the suggestion, our manuscript has been carefully revised in accordance with the above suggestions. At the same time, we have carefully improved discussion of the mechanisms involved in the comparative literature. Thank you very much for this proposal to make the overall language of our manuscript more rigorous and fluent.
Q4: Line 51-“First of all, MTB invade macrophages, which normally present antigen to T lymphocytes to release lymphokines that activate macrophages through positive feedback, enhance their maturation and acidification, and accelerate the mechanism for killing intracellular pathogens in autophagic lysosomes. However, MTB escapes the immune response by inhibiting these mechanisms.” Please cite appropriate references (Shea and Siegel Clinical Immunology, Third Edition. PMID: 28137237, 27763255 ). Does the author mean Cytokine or lymphokine here ? Please be consistent with the terms.
A4: We agree. First of all, we have added the appropriate references. Then, regarding the matching of cytokines and lymphokines with the terms, we changed the text. See revision. This better paves the way for the following explanation of this mechanism.
Q5: The role of PknG is missing in section on maturation of phagolysosomes (PMID: 23084287, 26359969, 15155913)
A5: Thanks. We have added new material on this topic.
Q6: Line 69 “The maturation of phagosomes can be arrested directly by live BCG vaccine, as phagosomes exibit only small amounts of the lysosomal glycoprotein LAMP-1 and processing of the lysosomal hydrolase cathepsin D is blocked)” This sentence is not clear.
A6: Your suggestion is of great importance to us. It makes the language of our manuscript more fluent. We have corrected this sentence. See revision.
Q7: Line 87 “Via a mechanism that activates host cytosolic phospholipase A2, these bacteria can promptly escape from the phagosome and create favorable conditions for survival in the cytoplasm of the host cell [17].” This sentence is not clear.
A7: Thanks for your suggestion, we have changed the sentence on the basis of references. See revision.
Q8:“Wag31, an MTB protein that belongs to the DivIVA family, protects MTB from oxidative stress induced cleavage due to its interaction with penicillin-binding protein through amino-acid residues in the nuclear receptor-binding SET domain [25].” The sentence is not clear
A8: We have changed the sentence. See revision. Based on your suggestion, we rethink the mechanism and organize our language. Thank you very much for your earnest guidance.
Q9:lent plasmid of Mycobacterium ulcerans (PMID: 25412098). This protein may helThe authors mention about Wag31 role. Interestingly a DivIVA homolog Mup012c is located on virup in protecting M. ulcerans during oxidative stress. I think this can be mentioned.
A9: Following your suggestion this topic has been added .The reviewer's comments give us a deeper understanding of the immune escape mechanism of Mycobacterium tuberculosis.
Q10:Line 199 “The P19 lipoprotein on the cell wall of weakly virulent MTB stimulates the differentiation of CD4+ cells and the release of cytokines like IL-6, IFN-γ, and IL-12”. Does the author mean CD4 cells secrete IL-12 ? Please provide reference and rewrite the sentence.
A10:Thanks. We have mporoved the text and added relevant references. See revision. Thank you again for your patient guidance.
Q11:Line 240 “This state is the most effective for killing parasitic pathogens, and the result is inhibition of phagosome-lysosome fusion[80] Please check. “ Reference is not correct.
A11: Thanks for your guidance. We have reviewed the reference and corrected it. Thanks for your timely reminder.
Q12: Figure 3 legend:and this action occurs in the plasma memrane However,
A12:Thanks for your guidance. We have corrected it.
Q13:Line 156. “ergothioneine (ERG). Gamma-glutamylcysteine (GGC) is an intermediate in glutathione and ERG synthesis. Five genes (egtA, egtB, egtC, egtD, and egtE) are involved in the synthesis of ERG;” Please see word fonts.
A13: In this section, we have carefully examined the fonts and corrected them. Thank you for your suggestions.
Q14:Line 168 “apoptosis of macrophages. .This indicated that factors, and interleukins (IL-6, -12, -4, and -10)” Please see formatting.
A14: Following your suggestion, we checked the format, found errors, and corrected them. See revision.
Q15:Line 173 “DCS and macrophages are the main components of the first line of defense against MTB, “Please see formatting.
A15: Based on your suggestion, we checked our format, found errors, and corrected them. See revision.
Round 2
Reviewer 1 Report
Thank you for the modifications
Author Response
We thank reviewer's response. We have also made more modifications for better qualification of our ms.
Reviewer 2 Report
The authors have done commendable job in answering reviewers concern. However, the manuscript still requires editing in English language since in many places it is difficult to understand the meaning. The authors only modified sentences where I pointed out. They need to make this manuscript understandable for average reader. I will cite few examples again for their convenience but the problem is throughout.
1.Natural killer T cells have an effect on resistance to MTB. I do not understand the meaning of this sentence.
2. "Via a mechanism...."
3. "KnG secreted by MTB prevents"
Author Response
We thank reviewer's helpful comments:
We have checked the whole manuscript by ourselves as well as English native speaker expert on the language and we polished the ms again.
According to reviewer's comments, we modified and highlighted the three parts in yellow and we ourselves made corrections in blue.
We are happy to improve our work for future publication.